# Calibrating Mini-Mental State Examination Scores to Predict Misdiagnosed Dementia Patients

Akhilesh Vyas [1,*], Fotis Aisopos [2], Maria-Esther Vidal [1,3], Peter Garrard [4] and George Paliouras [2]

1 L3S Research Center, Leibniz University of Hannover, 30167 Hannover, Germany; Maria.Vidal@tib.eu
2 National Centre for Scientific Research "Demokritos", Institute of Informatics & Telecommunications, 15341 Athens, Greece; fotis.aisopos@iit.demokritos.gr (F.A.); paliourg@iit.demokritos.gr (G.P.)
3 TIB-Leibniz Information for Centre for Science and Technology, 30167 Hannover, Germany
4 Neuroscience Research Centre, Molecular and Clinical Sciences Research Institute, St George's University of London , London SW17 0RE, UK; pgarrard@sgul.ac.uk
* Correspondence: akhilesh.vyas@tib.eu

**Abstract:** Mini-Mental State Examination (MMSE) is used as a diagnostic test for dementia to screen a patient's cognitive assessment and disease severity. However, these examinations are often inaccurate and unreliable either due to human error or due to patients' physical disability to correctly interpret the questions as well as motor deficit. Erroneous data may lead to a wrong assessment of a specific patient. Therefore, other clinical factors (e.g., gender and comorbidities) existing in electronic health records, can also play a significant role, while reporting her examination results. This work considers various clinical attributes of dementia patients to accurately determine their cognitive status in terms of the Mini-Mental State Examination (MMSE) Score. We employ machine learning models to calibrate MMSE score and classify the correctness of diagnosis among patients, in order to assist clinicians in a better understanding of the progression of cognitive impairment and subsequent treatment. For this purpose, we utilize a curated real-world ageing study data. A random forest prediction model is employed to estimate the Mini-Mental State Examination score, related to the diagnostic classification of patients.This model uses various clinical attributes to provide accurate MMSE predictions, succeeding in correcting an important percentage of cases that contain previously identified miscalculated scores in our dataset. Furthermore, we provide an effective classification mechanism for automatically identifying patient episodes with inaccurate MMSE values with high confidence. These tools can be combined to assist clinicians in automatically finding episodes within patient medical records where the MMSE score is probably miscalculated and estimating what the correct value should be. This provides valuable support in the decision making process for diagnosing potential dementia patients.

**Keywords:** dementia; mini mental score examination; machine learning; classification; regression; random forest; predictive models



## 1. Introduction

There are more than 8.7 million people across Europe living with dementia (Mapping-dementia-friendly-communities-across-europe https://ec.europa.eu/eip/ageing/library/mapping-dementia-friendly-communities-across-europe_en.html (accessed on 28 July 2021)). With an ageing population and no effective treatment, this number is set to rise to 152 million globally by 2050 (Dementia-number-of-people-affected-to-triple-in-next-30-years https://www.who.int/news/item/07-12-2017-dementia-number-of-people-affected-to-triple-in-next-30-years (accessed on 28 July 2021)), highlighting the huge unmet need for better managing this condition [1]. The sheer number of people living with the disease and the direct and indirect costs of providing care and support for them have made dementia quite challenging.

The Alzheimer's disease (AD) and the other neurodegenerative causes of late-life dementia have become one of the greatest medical challenges of our time. Diagnosis is quite challenging; it can be difficult to make and in a significant proportion of cases will be inaccurate. It is common for a diagnosis to be delayed for 2–3 years after symptom onset [2], and the presence of AD pathology will, on average, prove to be mistaken at postmortem examination, in as many as 25% of cases [3]. Currently, there are no licensed treatments that will stop, let alone reverse, the progression of the neurodegenerative pathologies. In recent years, however, understanding aspects of the biological and clinical characteristics of dementias has benefited from the ability to interrogate the increasing quantities of electronic data that exist for this growing patient population.

One example concerns the delineation of the typical pattern of response to one of the symptomatic treatments (cholinesterase inhibitors) that are licensed for the treatment of AD, using clinical evaluations recorded in thousands of electronic patient records. These evaluations included a summary estimate of a patient's cognitive ability in the form of a score (out of 30) on a bedside or office-based cognitive test such as the Mini-Mental State Examination (MMSE) [4] or the Montreal Cognitive Assessment (MoCA) [5].

Given the progressive nature of AD, it can be assumed that the score achieved by a given patient will remain unchanged for a variable length of time after the diagnosis has been made, and then gradually worsen. When the latter occurs, the score obtained at time $t$ will usually be lower than the score at time $t-1$. Other than in patients who enjoy a particularly strong therapeutic effect from cognitively-enhancing drugs such as acetylcholinesterase inhibitors [6], it is almost never the case that a patient's score on the MMSE undergoes a significant increase between two assessments. Moreover, the longer the interval between assessments, it is negative change rather than stability that is the expected pattern of change. Perera et al. [7] found that, other than in patients who started with MMSE scores of 10 or less, the improvement in MMSE score after starting a cholinesterase inhibitor seldom exceeded 5 points, and that the trajectory returned to progressive decline after as little as six months of treatment.

While knowledge of these trends may be helpful to individual clinicians when asked for a prognosis, and to health economists, epidemiologists and policymakers when considering the burden of the disease on society, their applicability depends crucially on the accuracy of the recorded MMSE data. Such an assumption would, however, be unsafe given the tendency for such clinical instruments to be used and scored in different ways by different clinicians, not to mention for accidentally erroneous values to be entered in the record. Therefore, a challenge that arises in the interpretation of big data such as electronic patient records (EPR) is the identification of clinical episodes where the MMSE score has been incorrectly calculated or inaccurately recorded. Previous work [8] supports the assumption of a set of baseline rules for better identifying diagnostic inaccuracies, based on the differences between MMSE scores obtained on consecutive episodes of assessment. A more advanced approach, however, could attempt to identify an inaccurate MMSE using features associated with an individual episode. A further challenge is to also investigate whether the MMSE score values can be correctly and effectively predicted using information from electronic health records.

This paper employs machine learning models to attempt an accurate MMSE score prediction, based on a set of clinical, molecular (e.g., APOE type), and demographic features associated with each patient episode. The main aim of this prediction is to determine whether instances of specious and/or inconsistent documentation of MMSE scores, which are frequently documented in clinical records, can be identified and corrected. In the setting of the current experiments, a patient's MMSE score was defined as erroneous if one of a number of conditions are met. These conditions, based on a priori assumptions derived from the clinical experience of one of the authors (PG), were as follows: if the change in MMSE value between two consecutive episodes of testing either (a) increased by more than five points during any interval up to a year, (b) increased by three or more points between the first and second years after diagnosis, or (c) increased by any number of points

over an interval greater than 2 years, an erroneous MMSE score had been documented. The solution to the aforementioned problem entails two steps. First, these rules are used to isolate the assessments at which erroneous MMSE scores are recorded from those in which the value of the MMSE score is plausible. Secondly, to train and test an MMSE score prediction model to determine what proportion of miscalculated cases are replaced by a predicted MMSE value, and what proportion of the scores presumed to be correct are left unchanged by the prediction model.

We hypothesise that the MMSE test and other clinical factors reported in electronic health records represent rich sources of information for cognitive assessment and AD severity. Built on this hypothesis, the predictive capacity of machine learning algorithms is exploited to forecast the MMSE value of a patient in a clinical episode and to classify if the patient is misdiagnosed or not. The accuracy of these models is experimentally assessed, and the observed results provide evidence of the importance that the clinical conditions have on the AD assessment. Further clinical interventions are required to validate the assumptions that guided our work. Nevertheless, the analytical validation based on traditional metrics that measure the accuracy of the predictive models, indicates that these results are promising and strengthen the premise that a holistic evaluation of a patient is required for an accurate diagnosis. Although several approaches in the literature report the use of machine learning on combined datasets that include MMSE test and other clinical factors [3,9–11], to the best of our knowledge, the predictive problems stated in this paper have not modelled with machine learning models trained over the OPTIMA dataset [2,12,13]. Thus, this work not only evidences the role of machine learning, but represents a novel contribution to the portfolio of accurate tools to support clinicians in the challenging task of diagnosing dementia patients.

The principal stages of this work are summarised as below:

- Cleaning and curating data from a real-world ageing study, based on certain rules, to test and validate meaningful predictions.
- A prediction (Regression and Classification) model that provides MMSE score estimations for patient episodes, based on various features of those episodes, and identifies the most important features for calculating the MMSE value.
- A prediction model that classifies patient episodes with erroneous MMSE scores, based on other clinical features associated with the episode.

The paper is structured as follows: the current Section 1 introduces the background and motivation of the work. Section 2 presents recent related work on dementia diagnostic models and MMSE score predictions, while Section 3 formulates the problem addressed and analyses the methods employed in this work. Lastly, Section 4 presents the OPTIMA dataset [2,12,13], cleaning and curation process, the experimental results and their evaluation on our dataset, Section 5 covers discussion about our approach and Section 6 concludes the current and future work.

## 2. Literature Review

### 2.1. Mini-Mental State Examination

The MMSE was first introduced in 1975 [4] as a clinical approach to quantifying the cognitive status of patients with suspected neuro-degenerative dementias. Its diagnostic properties have been replicated in a number of studies, both in its original English format and after adaptations for use in other languages. It has since been adopted as a routine component of the cognitive assessment of patients with Mild Cognitive Impairment (MCI) or AD, in the context of both clinical evaluation and research studies. A rival method of screening for and evaluating patients with dementia is the Clock Drawing Test (CDT) [14]. The CDT provides a simple scoring system for the rapid screening of cognitive impairment in patients with MCI and Dementia [15].

The MMSE has also become the most popular and widely used screening instrument in dementia [10], with the cutoff of 26 being defined as distinguishing cognitively normal people from those with incipient or established dementia. A value of 27 is used in

subpopulations with high levels of education [16]. The MMSE can also be used to delineate the different stages of AD, with a score of 20 to 24 suggesting mild dementia, 13 to 20 suggesting moderate dementia, and less than 12 indicating severe [17].

The MMSE has good sensitivity and specificity across all age and educational groups [18]. In contrast, the Montreal Cognitive Assessment (MoCA), an alternative brief cognitive screening tool, which is also scored out of 30, may be more sensitive at discriminating MCI and Alzheimer's Disease, but has lower specificity, misclassifying more cognitively normal elderly controls than the MMSE [5]. MMSE has been also validated in dementia other than AD—e.g., dementia with Lewy Bodies and Parkinson's disease dementia [19]—and has neurophysiological correlates: a recent study [20] found that a combination of the peak latency of the p300 component of an event-related potential, and the power associated with the alpha frequency of the electroencephalogram, together with age and years of education, were able to predict the MMSE score with an accuracy of three points. Arevalo-Rodriguez et al. [8] studied the role of MMSE as a predictor of which patients diagnosed with the prodromal syndrome of MCI would go on to develop dementia. Although a one-off test score is unlikely to have much predictive power to answer this question, monitoring the rate of change over time may be more informative, suggesting that the role of repeated MMSE measurements could prove more informative than single baseline scores. A review by Creavin et al. [21] also attempted to determine the diagnostic accuracy of MMSE at various cut points for dementia in people aged 65 years and over. The authors concluded that MMSE contributes to a diagnosis of dementia in low prevalence settings, but should not be used in isolation to confirm or exclude disease. It was also proposed that it would be important for future investigators to understand whether administering the MMSE changes patient-relevant outcomes.

Most of the above work will have been conducted prospectively, using dedicated test protocols carried out by trained individuals, and therefore not likely to contain inaccuracies. However, the power and versatility of the MMSE is such that values recorded in electronic patient records, and therefore available as a component of big data resources, could be exploited to interrogate a variety of clinical and biological questions from a fresh perspective. Such values are, however, typically obtained during clinical visits, often under time pressure, and by individuals with different approaches to testing and scoring, and therefore may be subject to variability and/or inaccuracy.

In this paper, we report work investigating serial MMSE tests in a cohort of patients at various stages of dementia, in order to identify instances of misdiagnosis in the form of MMSE score miscalculations. We use the patients' histories, along with lifestyle, demographic and other features [3] to build machine learning classification models to predict the MMSE scores and correct the 'misdiagnosed' cases, without significantly affecting the values recorded in the correct episodes.

### 2.2. Cognitive Assessment Using Machine Learning

There are a number of current controversies in machine learning (ML) that may be relevant to the present work. According to Flaxman and Vos [22], developing machine learning approaches requires a broad range of experience and clinical correlation. ML techniques remain constrained by issues of fairness, accountability, transparency, privacy, explainability, and causal inference. Finally, the performance of ML models is mostly influenced by the dataset's size and heterogeneity [23].

Based on the Korean Dementia Screening Questionnaire (KDSQ) and the MMSE, a machine learning algorithm (logistic regression) was used to diagnose cognitive impairment utilizing 24 variables including education, sex, age, and hypertension [11]. The MMSE-based model predicted cognitive impairment better than the KDSQ-based model. Aram So et al. [9] employed neuropsychological and demographic data to predict normal, mild cognitive impairment (MCI) and dementia using naive Bayes, Bayes network, bagging, logistic regression, random forest, support vector machine (SVM), and multi-layer perceptron (MLP) machine learning classification models. They claimed that it improved existing

clinical practice by quickly, inexpensively, and reliably detecting dementia at an early stage, as well as increasing screening accuracy. In this paper, we review machine learning models that are used to handle classification problems involving cognitive impairment utilising clinical features.

### 2.3. Random Forest

The random forest model has received a lot of attention in recent years for its effectiveness on classification and regression issues in a variety of scientific domains (e.g., bioinformatics, genetics) [24–26]. This model outperforms other supervised machine learning models when dealing with overfitting, noisy non-linear, high-dimensional, and multi-modal data (e.g., SVM, logistic regression, naive Bayes) [27].

Velazquez et al. [28] applied a random forest model to predict early conversion of mild cognitive impairment (MCI) to Alzheimer's Disease (AD) based on clinical features. They also adopted oversampling to resolve class imbalance before training and achieved high accuracy (93.6%) with a random forest model for the prediction task. Because the random forest model was able to distinguish between both groups, it outperformed the other classification model, SVM. In another machine learning classification task, random forest algorithm was successfully applied on multi-modal neuroimaging data for the prediction of Alzheimer's disease instead of using other supervised machine learning algorithms [29]. It demonstrated how well the random forest model performed on high-dimensional, multi-modal data, and it was able to collect complementary information across modalities. We tested a variety of classification models, but the random forest model surpassed other machine learning models empirically.

### 3. Problem & Approach

In this section, we define the MMSE miscalculation problem and the approaches taken to address it.

### 3.1. Problem Statement

The Mini-Mental State Examination (MMSE) is used in clinical research studies to assess the degree of cognitive impairment [8] that has resulted at a given point in time from a dementia-causing condition. It is expressed as a score, with a maximum of 30, based on a patient's ability to answer questions that make demands on different categories of cognitive ability, such as orientation in time and place, use of words, attention, calculation, and both short term and long term memory [30]. The score achieved on the MMSE has been conventionally used to place a patient in one of four categories of cognitive ability, namely: cognitively normal, mild dementia, moderate dementia and severe dementia [4]. In a patient with dementia due to the Alzheimer's disease, the score becomes progressively lower as the disease progresses [31]. However, the accuracy with which a patient's MMSE measurement is either measured, recorded or documented, is compromised. Such inaccuracies may occur for multiple reasons, including the patient being without hearing aids or glasses, a transient motor problem, low motivation, or even an inadequate rapport with the administering clinician [32]. Miscalculations become evident when a patient's MMSE is recorded serially, and an unexpected increase or decrease in the recorded MMSE score.

By searching for instances in which one of the a priori criteria for a miscalculation is met, MMSE scores may be classified as being either inaccurate ("misdiagnosed = YES") or accurate ("misdiagnosed = NO"). This classification indicates whether the patients have been correctly diagnosed, in the sense of whether the MMSE score has been miscalculated or erroneously recorded in any of their documented assessments. In our case, the problem we aim to address is twofold: (a) MMSE Prediction: use those features to predict the MMSE scores for each patient episode, in order to correct misdiagnosed cases (by replacing miscalculated scores with more reasonable predicted values). (b) Classification problem: predict the class ("misdiagnosed = YES"/"misdiagnosed = NO") of patients using patient

clinical record features including MMSE features from the OPTIMA dataset [2,12,13]. Machine learning predictive models address these problems.

### 3.2. Approach and Metrics

#### 3.2.1. Baseline-MMSE Calculation Using Rule-Based Approach

To address the problems defined in Section 3.1 for a set of patient medical records, we must first determine a baseline for misdiagnosis. In the present study, this was done using a hand-crafted rule that is based on the experience and observations of the medical expert (PG) in the course of assessing and treating patients with similar conditions. The hand-crafted rule is based on the identification of MMSE score changes over time that are likely to indicate a misdiagnosis. The rule is defined as follows:

A patient is designated as being in the "**misdiagnosed = YES**" class if the change in two consecutive MMSE values of the patient's episodes satisfies one of the following conditions:

1.  MMSE-CHANGE $> 5$ in Episode-Interval $\leq 1$ year;
2.  MMSE-CHANGE $\geq 3$ in Episode-Interval $(1, 2)$ year;
3.  MMSE-CHANGE $\geq 0$ in Episode-Interval $\geq 2$ years.

If none of the above conditions obtains, the patient falls into the **"misdiagnosed = NO"** class. Albeit empirically defined, this rule is supported on the clinical experience.

#### 3.2.2. Predictive Models

Our goal is to improve the baseline accuracy of 'misdiagnosis', occurring due to the imprecise nature of the MMSE measurements, by employing machine learning-based predictive models. In supervised machine learning, a decision tree algorithm is widely used as a predictive modelling approach due to its explainability. The decision tree algorithm (such as ID3 [33], C4.5 [34] and CART [35]) is a white box method which learns the decision rules from data features and predicts the value of the target variable [33]. However, very complex trees (due to complex decision rules) do not generalize the data well and tend to overfit. In addition, individual decision trees suffer from high variance [36].

We also perform experiments with other supervised machine learning models such as SVM, MLP and logistic regression. Nevertheless, we keep our focus on the random forest model (advantages and applications are mentioned in the literature review Section 2) for experiments due to unsatisfactory results from other models. Our all predictive models implement an ensemble learning method implemented by the random forest model, which utilizes different decision trees over sub-samples of the dataset [37]. The decision of the algorithm uses the average of results provided by different decision trees. Besides, using the decision tree algorithm over a sub-sample of a dataset helps random forest to control over-fitting [36]. In our experiments, we make use of two predictive modelling approaches to deal with the high imbalance of the classes. We discuss in detail how the first model utilizes the majority class and the problems associated with the second predictive model due to class imbalance. The predictive models are described as follows:

*   Regression and Classification Model: The regression model attempts to predict the MMSE value of a patient's episode, based on various input features, provided in Listing A1. The regression model is trained using an ensemble learning of random forest Regressor. Then, the difference between predicted MMSE value and previous episode MMSE value in the given episode-interval is used to decide the class of each patient (misdiagnosed = 'YES'/'NO'), as mentioned in the hand-crafted rules of Section 3.2.1.
*   Classification Model: The random forest classification model simply attempts to predict the class (misdiagnosed = 'YES'/'NO') of patients, based on the various input features, provided in Listing A1. We also deal with the imbalance of the two classes in the dataset to avoid classification error towards minority class.

### 3.3. Evaluation Metrics

Accuracy is a heuristic statistical performance measure [38] for the classification problems. It is simply a ratio between correctly predicted instances divided by the total number of instances. However, it does not reflect the actual performance of the model when there is an uneven class distribution. In addition, it does not give detailed information regarding the performance of the model to the problem. Therefore, to measure the actual performance (validity of results and completeness of results) of the predictive models, we choose all performance measures precision, recall, f1-score, and accuracy. When evaluating classification models, the precision of a class is the number of true positive predictions, divided by the total number of records predicted as belonging to this class, representing a kind of "confidence" for predictions of this class. A recall is the number of true positive predictions divided by the real number of records of this class, representing the model's ability to find all records of the class. The metric of f1-score provides a weighted average of precision and recall to seek balance between precision and recall. It takes both false positives and false negatives into the account.

For example, when classifying patients in the "misdiagnosed = YES" class described in Section 3.2.1, precision is the ratio of correctly predicted "misdiagnosed = YES" class patients episodes to the total patient's episodes predicated as "misdiagnosed = YES". On the other hand, recall is the ratio between correctly predicted "misdiagnosed = YES" class patients episodes to all patients episodes labelled as the "misdiagnosed = YES" class originally. We repeat the same measures for patient's episodes classified in the "misdiagnosed = NO" class.

## 4. Experiments and Results Analysis

In this section, we explain the experimental settings of the predictive models, and we describe the dataset employed along with the cleaning and pre-processing steps.

### 4.1. Dataset

The OPTIMA [2,12,13] project, which was active from 1988 until 2008, was a longitudinal cohort study of ageing and dementia, recruiting individuals aged over 70 with either normal or mildly impaired cognition, and taking annual measurements of physical, metabolic, imaging, clinical, and cognitive indices until death. Most participants agreed to allow their brains to be examined post-mortem to confirm the cause of the underlying dementia (if present) or the absence of pathological change (if no dementia). The aim was to gain insights into the factors predisposing to dementia, in order to optimise preventive strategies and guide the development of new treatments. The OPTIMA dataset includes, for each patient, a medical history, results from physical examinations, neuropsychological assessments, blood test results relevant to the major organs, vitamin levels and other metabolic indices, and brain scans (CT, MRI, and fMRI). The OPTIMA dataset, used in this study, was received in January 2019. It consists of 1035 distinct patients with 9584 recorded episodes and their features. Moreover, the dataset contains unique patient IDs, with each ID and episode date allowing unique identification of a each assessment of a given patient's condition over time. Each episode is linked to 1593 different features of the types outlined above. The medical observations at each visit cover the patient's health and well-being, the findings of cognitive and physical exams, and any concurrent medical (e.g., cardiovascular) conditions. In addition, the patients' MMSE scores are presented in sequential episodes to quantify their cognitive assessment in a principled manner. On the basis of qualitative and quantitative analysis, as well as the importance of attributes (various related works for selecting features [3]), we selected the most relevant features to train our predictive models.

Attributes with large numbers of missing values can be filled using relevant attributes from other episodes. For example, height and weight are used to replace missing BMI feature values. Some attributes values are life-long and cannot be altered in other patient episodes, so they are also filled using values for past (or future) episodes. For instance,

the gender attribute of a patient is more constant throughout life generally, and can therefore be filled in all cases using a past or future episode record of the patient. Moreover, attribute values that do not show an unexpected change over a period during a patient's history can be also filled using values recorded in other episodes. Previous studies have shown that demographics, present fitness status, life-style, education-level, social-life and working conditions play a significant role in cognitive diseases [3]. We therefore include these attributes in the training of our predictive model, before data cleaning and pre-processing. Attributes with a notable number of missing values are removed from the attribute list.

*4.2. Dataset Cleaning and Preprocessing*

As a preliminary step, we eliminate all the patients' episodes from the dataset whose MMSE values are outside the standard MMSE examination score range [0, 30]. Second, we only include patients who have been diagnosed with MCI in one of their first episodes (this can be determined using the feature "Petersen MCI==1" in each episode). It serves as a starting point for calculating episode intervals in the patient's episode classification using hand-crafted rules. Only 1104 episodes from the patients can be categorised when all the facts are taken into account. In detail, patients episodes are classified into two classes by utilizing the predefined hand-crafted rule of Section 3.2.1, "misdiagnosed = YES" (971 patients) and "misdiagnosed = NO" (133 patients) out of 1104 episodes. Missing values in the features of the patients are imputed as described in Section 4.1. We transform ordered categorical features by applying ordinal encoding (e.g., size as Small, Medium, Large to integer values 0, 1 and 2, respectively), non-ordinal categorical features by utilizing one-hot encoding (https://scikit-learn.org/stable/modules/preprocessing.html (accessed on 28 July 2021)) and keep numerical features in their original form. Finally, we perform model training with the list of selected features provided in the Listing A1.

*4.3. Experiments*

In this section, we present the experimental settings and evaluate the results of the predictive models for each problem. Problems (a) and (b) are addressed using the two supervised machine learning models introduced above: a regression and classification model, and a simple binary classification model.

4.3.1. Regression and Classification Model

Initially, we train our regression model only using "misdiagnosed = NO" class data instances to be able to efficiently predict the MMSE score values. To optimise the model testing process on balanced classes ("misdiagnosed = "NO"/"YES""), we split the "misdiagnosed = NO" cases set into 5 folds. Each time, 4 different folds are used for training and the remaining one is merged with 133 data points of "misdiagnosed = YES" class for testing. This procedure is repeated five times, considering a different fold of "misdiagnosed = NO" class with the 133 "misdiagnosed = YES" class cases each time. This dictates that the regression will be tested using all possible cases of "misdiagnosed = NO" class, in order to ensure the data is thoroughly examined. After each repetition, predicted MMSE values of the data points are used to instantiate the baseline rule by experts (Section 3.2.1) and re-assess their classes ("misdiagnosed = "YES"/"NO""). The performance of the predictive model is summarized in Tables 1 and 2, in terms of classification results (based on the original classification of MMSE scores) and confusion matrices, respectively. The average of all obtained matrices after five training repetitions is used to compute the average classification report and confusion matrix.

Result Analysis

As explained, the main objective of this model is to produce accurate MMSE score predictions that can be used to replace previously miscalculated MMSE values in "misdiagnosed = YES" cases. However, since the predicted scores must be applied in correctly diagnosed episodes as well, they should approach the correct MMSE values, in order not

to seriously affect the new classification of those cases. Thus, we first train the regression model, in order to learn the approximate function to predict "misdiagnosed = NO" patient's MMSE values, and thus, minimise the regression error for "misdiagnosed = NO" and maximise for the "misdiagnosed = YES" patients.

Table 1 demonstrates that for originally "misdiagnosed=YES" cases, the resulting classification has a low recall and significantly high precision in average for all folds. The low recall score means that the regression model manages to "correct" more than half of the 133 originally misdiagnosed cases (77.6 Original:YES/Predicted:NO in average) as shown in Table 2. Thus, an significant percentage of previously misdiagnosed episodes, now have a more realistic MMSE score provided by our model.

**Table 1.** Classification report after 5 repetition of model training and testing.

| | Average Classification Report | | | |
|---|---|---|---|---|
| | **Precision** | **Recall** | **f1-Score** | **Support** |
| NO | 0.71 | 0.98 | 0.82 | 194.2 |
| YES | 0.93 | 0.42 | 0.58 | 133 |
| accuracy | | | 0.75 | 327.2 |
| macro avg | 0.82 | 0.70 | 0.70 | 327.2 |
| weighted avg | 0.80 | 0.75 | 0.72 | 327.2 |

**Table 2.** Average confusion matrix.

| | | Predicted Class | |
|---|---|---|---|
| | | **YES** | **NO** |
| **Original Class** | **YES** | Orig:YES Pred:YES 55.4 | Orig:YES Pred:NO 77.6 |
| | **NO** | Orig:NO Pred:YES 4.2 | Orig:NO Pred:NO 190 |

On the other hand, only a few originally correct cases out of 133 are now classified as misdiagnosed, based on the predicted MMSE scores (4.2 Original:NO/Predicted:YES on average). This is also demonstrated in the high recall of the "misdiagnosed = NO" cases. This observation means that 98% of the correct diagnosis cases did not turn into "misdiagnosed = YES" using the predicted MMSE scores. Therefore, the model has managed to correct a significant number of previously "misdiagnosed = YES", whilst not seriously affecting the "misdiagnosed = NO" class.

Important Features

Many attributes may impact the course of the predictive model. However, we illustrate the most important features in Table 3. As expected, most of these variables are related to individual aspects of the MMSE examination, and therefore, directly influences the prediction of MMSE value. We also illustrate other important features in Table 4 that are not related to the MMSE score questionnaire. These are also important, as they are the parameters highly contributing to the MMSE score estimation, without being directly part of the examination.

4.3.2. Classification Model

This predictive model is implemented using a random forest classifier with the default parameter 5-fold-cross validation. In each iteration (total five iterations), the model is trained on 4 folds of the data and tested on 1 fold. Finally, we merge all predicted classes

to return the final result. In this series of experiments, we decided to employ oversampling [39] and undersampling [40] algorithms for solving the imbalance between the two classes of the dataset, to avoid unwanted side effects on the classification results. Since most of the classification algorithms are designed around the assumption of an equal number of instances for each class, imbalance may cause these models to be overwhelmed by the majority class and ignore the minority class.

**Table 3.** Important Feature list for identifying misdiagnosed patients.

| |
|---|
| COGNITIVE EXAM: IDENTIFIES DAY OF WEEK |
| COGNITIVE EXAM: IDENTIFIES MONTH |
| COGNITIVE EXAM: IDENTIFIES YEAR |
| COGNITIVE EXAM: IDENTIFIES COUNTY |
| COGNITIVE EXAM: COMPREHENDS RADIO |
| COGNITIVE EXAM: NUMBER OF ANIMALS LISTED |
| COGNITIVE EXAM: NUMBER OF ANIMALS LISTED: SCORE |
| COGNITIVE EXAM: REGISTERS OBJECTS 3: PENNY |
| COGNITIVE EXAM: RECALLS OBJECTS |
| COGNITIVE EXAM: RECALLS OBJECTS 1: APPLE |
| COGNITIVE EXAM: WRITES A SENTENCE |
| COGNITIVE EXAM: PRAXIS PAPER |
| COGNITIVE EXAM: PRAXIS PAPER: ON LAP |

**Table 4.** Important Feature list for identifying misdiagnosed patients other than MMSE related questions.

| |
|---|
| ANXIETY/PHOBIC |
| CAMCOG REMOTE MEMORY SCORE |
| CERBRO VASCULAR DISEASE PRESENT |
| CLINICAL BACKGROUND: BODY MASS INDEX |
| DEPRESSIVE ILLNESS |
| DIAGNOSTIC CODE |
| EST OF SEVERITY OF DEPRESSION |
| EST SEVERITY OF DEMENTIA |
| PRESENT STATE: MEMORY PROBLEM |

Therefore, the evaluation of the model uses different distribution of classes by employing the following imbalance algorithms:

- Original-dataset: There are no changes in the dataset regarding the imbalance of the classes.
- Oversampled-dataset: It is oversampled by the minority class ("misdiagnosed = YES") data-points using the Synthetic Minority Oversampling Technique (SMOTE) [39]. The idea of the algorithm is to generate new data points for the minority class "misdiagnosed = YES" equal to the majority class "misdiagnosed = NO" for better evaluation of the model. However, this approach fails due to the very low minority class data-points.
- Undersampled-dataset: The majority class ("misdiagnosed = NO") data-points are undersampled using cluster-centroids algorithm [40]. The idea of the algorithm is to generate several clusters, calculated by the k-means algorithm, with each cluster containing similar data. Then, the original data in the same groups are replaced by the cluster centres, thereby reducing the size of the majority class ("misdiagnosed = NO"). The outcome of this process provides an equal distribution of classes for better evaluation of the model performance.
- Random-Undersampled-dataset: In this setting, the majority class ("misdiagnosed = NO") data-points are undersampled using random points generated by a random-sampler algorithm. The basic idea of the algorithm is to randomly sample data points for the

majority class "misdiagnosed = NO" to make it equal to the minority class "misdiagnosed = YES". The outcome of this process again provides an equal distribution of classes for better evaluation of the model performance.

Result Analysis

Table 5 illustrates the performance of all models in terms of precision, recall, f1-score, and accuracy.The imbalance of the classes in the dataset plays a significant role in deciding the performance of the predictive models. The model based on the original dataset suffers due to the high imbalance in "misdiagnosed = YES" and "misdiagnosed = NO" classes. It is not able to learn for "misdiagnosed = YES" class (133 data points) due to a very less number of data points compared to the "misdiagnosed = NO" class (971 data points) in the original dataset. The second (oversampled dataset) model performs better on the "misdiagnosed = YES" class compare to the model of original-dataset as a result of the dense cluster of generated data points for this class. The dense cluster is formed due to the high similarity between generated data points features or less distance between data points. Both datasets are affected by the imbalanced distribution of the classes. There is also a significant difference between the performance of predictive models using datasets (generated by cluster-centroids and random-under-sampler algorithms) in terms of f1-score as shown in Table 5.

**Table 5.** Classification report after different sampling of dataset.

| | Precision | Recall | f1-Score | Support |
|---|---|---|---|---|
| **Original Dataset** | | | | |
| NO | 0.88 | 0.99 | 0.93 | 971 |
| YES | 0.26 | 0.04 | 0.07 | 133 |
| accuracy | | | 0.87 | 1104 |
| macro avg | 0.57 | 0.51 | 0.50 | 1104 |
| weighted avg | 0.81 | 0.87 | 0.83 | 1104 |
| **Oversampled Dataset (SMOTE)** | | | | |
| NO | 0.93 | 0.96 | 0.94 | 971 |
| YES | 0.96 | 0.93 | 0.94 | 971 |
| accuracy | | | 0.94 | 1942 |
| macro avg | 0.94 | 0.94 | 0.94 | 1942 |
| weighted avg | 0.94 | 0.94 | 0.94 | 1942 |
| **Undersampled Dataset (ClusterCentroids)** | | | | |
| NO | 0.92 | 0.91 | 0.91 | 133 |
| YES | 0.91 | 0.92 | 0.91 | 133 |
| accuracy | | | 0.91 | 266 |
| macro avg | 0.91 | 0.91 | 0.91 | 266 |
| weighted avg | 0.91 | 0.91 | 0.91 | 266 |
| **Undersampled Dataset (RandomUnderSampler)** | | | | |
| NO | 0.70 | 0.74 | 0.72 | 133 |
| YES | 0.72 | 0.68 | 0.70 | 133 |
| accuracy | | | 0.71 | 266 |
| macro avg | 0.71 | 0.71 | 0.71 | 266 |
| weighted avg | 0.71 | 0.71 | 0.71 | 266 |

Clearly, not all models can accurately address the problem of discriminating episodes for both classes ("misdiagnosed = NO" and "misdiagnosed = YES"). The results show how the imbalance in the classes makes it difficult to select the right model, in order to identify "misdiagnosed = NO" patients among misdiagnosed-YES patients. The best models for the task at hand seem to be the one using the oversampled dataset (SMOTE) and the cluster-centroids undersampled (average f1 scores of 0.94 and 0.91, respectively), which can discriminate episodes effectively for both classes. This high f1-score indicates better precision and recall of the predictive models. Both models have a high number of true positives, less false positive & false negative for both classes.

Practically, the results above prove that our model, with the assumption that it has been trained with many cases for both classes, can accurately identify misclassified MMSE score cases in patient episodes. To do so, the training data for each episode has to include a rich number of patient-related parameters, including cognitive tests (such as CAMCOG), demographic, clinical and other variables.

## 5. Discussion

The experiments presented in this study illustrate the predictive value of demographic features combined with past measurements of specific aspects of a cognitive examination (the MMSE) in the effective prediction of a patient's score on the same instrument. The resulting MMSE predictions can therefore be utilised with the aim to better approach correct MMSE values, as many miscalculations may have been entered into an electronic database for various reasons. In this way, assessments of the cognitive state of an individual at each assessment episode become more reliable methods of distinguishing cognitively normal people from those with incipient or established dementia, and of estimating the severity of dementia that is present in the latter.

It is axiomatic that the performance and value of any analytical model will be limited by the quality of the available data [41]. Outside clinical circles, however, the potential for inaccuracy in medical reporting is perhaps less well recognised, and in the evaluation of patients with cognitive disorders—a process that is still largely driven by observation, experience that is sometimes, but not always, guided by the application of consensus criteria—disagreements and erroneous diagnoses are particularly rife. This is largely because the majority of cases of dementia are caused by the gradual accumulation of molecular level damage that has no effect on function, cognition or behaviour until relatively advanced [42] and was impossible to detect with certainty during life [43,44].

In machine learning models, dataset size and heterogeneity (e.g., diverse populations) are always important [23]. However, finding an adequate dataset in the healthcare industry is usually difficult. We need additional data and diverse data to solve generalisation issues because we train our predictive models on a small and old dataset. Many obstacles may arise while deploying our machine learning model models in real-world scenarios, such as differentially dispersed datasets, retraining, re-calibration, and generalizability. Technical differences, such as various measuring equipment, coding definitions, EHRs systems, medical personnel, and so on, are also predicted to cause this generalization problem. As a result, it is always important to record shifts in new cases to assess and improve the predictive models' effectiveness. One of the alternatives for this is data-driven testing [45].

Nonetheless, it is possible to envisage scenarios in which the approach described in this work may be of practical clinical use. One is in research involving electronic health records in which MMSE scores are recorded as part of the clinical assessment and is of interest to the researcher, whether as an outcome or a covariate. The researcher is likely to encounter at least as many (and in practice probably a lot more) erroneously calculated MMSE scores in such an unstructured dataset. The ability to detect between-group differences in values or the influence that the values may exert on another comparison, would be enhanced by having a dataset from which these erroneous values had been replaced by plausible ones. Another situation in which automatic detection of an anomalous MMSE score could be useful would be in the context of a clinical decision-support system which had learned

to detect and alert the clinician to a value of MMSE that was implausibly different from previously recorded scores. Such a system would contribute to the goal of accuracy in clinical record keeping, but would also be useful in the context of ensuring data quality in research settings (such as a pharmaceutical trial) where change in MMSE score was an outcome of interest.

Finally, the increasing availability of large datasets in the form of both open data resources (such as DPUK (https://www.dementiasplatform.uk/ (accessed on 28 July 2021)) or the AD & FTD Mutation Database (https://uantwerpen.vib.be/ADMutations (accessed on 28 July 2021))) and more recently described methods for integrating these different sources of information [46–48] is set to further improve the analytical and predictive capabilities of data science in complex domains. These include the interplay of factors that under the development of common but aetiologically heterogeneous medical conditions, including dementia [49]. While medical datasets that incorporate sufficient structured information are currently few and far between, such resources are increasing in number and availability due to the growing recognition of the potential of data science and data mining and the increasing number of relevant data portals. Important and potentially powerful examples of the latter include tools for interrogating and anonymously harvesting clinical information from aggregated EPRs. An early and influential example of this approach to clinical data assets without compromising data confidentiality has been the UK's Clinical Record Interactive Search (UK-CRIS) facility, which has delivered original insights into aspects of diagnosis and management of mental health disorders [50]. Because (at least in the UK) the diagnosis and management of dementia has historically been included within the purview of mental health professionals, similar insights have been possible in this increasingly important domain of clinical research [51,52].

## 6. Conclusions and Future Work

The analyses presented in this study provide preliminary evidence that machine learning approaches may be helpful in the task of optimising the accuracy of big data assets which, although potentially highly informative, are also vulnerable to the presence of inaccuracies. When in vivo diagnostic tools such as molecular ligands [53–55] become widely available, it will be possible to address and retrospectively to ameliorate diagnostic inaccuracies in large clinical dementia datasets. This will, in turn, lead to new insights into disease risks, mechanisms, the discovery of endophenotypes and patient stratification. Although we did not have access to diagnostic ground truth, the use of MMSE scores as a proxy variable is presented as evidence for the feasibility of using AI to improve diagnostic accuracy within large datasets through the implementation of ML algorithms such as RF.

One of the biggest strengths of this study was the availability of a large and comprehensive, structured data set containing values of numerous variables obtained from an ageing population. In contrast, much of the clinical data that are currently available from EPR are unstructured and require NLP pre-processing to extract a database of similar numerical values to those used her—a database that will be less comprehensive and likely to contain missing values. Since MMSE scores are both amenable to accurate extraction using NLP and widely documented in clinical assessments, replication of the current study on a larger and more ecologically valid database should be possible.

It will, in addition, be important to substitute data driven methods to create criteria for accuracy of recorded scores, as the 'handcrafted' criteria adopted here, while unlikely to overestimate the rate of deviation from scoring accuracy may have underestimated it. The development of methods for deriving accuracy criteria from the characteristics of the data themselves, and their implementation are among the studies currently in progress in our consortium, and we expect to report on them in the near future.

**Author Contributions:** Conceptualization: A.V. and P.G.; methodology: A.V.; software: A.V.; validation: A.V. and P.G.; formal analysis: A.V., F.A., M.-E.V. and P.G.; investigation: A.V., F.A., P.G. and M.-E.V.; resources: M.-E.V., P.G. and G.P.; data curation: A.V.; writing—original draft preparation: A.V., F.A. and P.G.; writing—review and editing: A.V., F.A., M.-E.V. and P.G.; supervision: M.-E.V.

and P.G.; project administration: G.P.; funding acquisition, M.-E.V. and G.P. All authors have read and agreed to the published version of the manuscript.

**Funding:** This paper is supported by European Union's Horizon 2020 research and innovation programme under grant agreement No. 727658, project IASIS (Integration and analysis of heterogeneous big data for precision medicine and suggested treatments for different types of patients).

**Institutional Review Board Statement:** Not applicable.

**Informed Consent Statement:** Not applicable.

**Data Availability Statement:** The data that support the findings of this study were provided from the Oxford Project to Investigate Memory and Ageing (OPTIMA) via a bilateral agreement with the IASIS project and cannot be publicly shared. The source code of models is available at https://github.com/SDM-TIB/dementia_mmse.git (accessed on 28 July 2021).

**Acknowledgments:** The authors would like to acknowledge the OPTIMA project for providing access to their clinical dataset.

**Conflicts of Interest:** The authors declare no conflict of interest.

## Abbreviations

The following abbreviations are used in this manuscript:

| | |
|---|---|
| OPTIMA | Oxford Project to Investigate Memory and Ageing |
| MMSE | Mini-Mental State Examination |
| AD | Alzheimer's disease |
| MoCA | Montreal Cognitive Assessment |
| EPR | Electronic Patient Record |
| MCI | Mild Cognitive Impairment |
| CDT | Clock Drawing Test |
| SMOTE | Synthetic Minority Oversampling Technique |
| CT | Computed Tomography |
| fMRI | Functional Magnetic Resonance Imaging |
| MRI | Magnetic Resonance Imaging |
| MCI | Mild Cognitive Impairment |
| RF | Random Forest |
| NLP | Natural Language Processing |

## Appendix A

**Listing A1.** Complete list of selected attributes

```
[ 'CLINICAL BACKGROUND: BODY MASS INDEX' , 'ANXIETY/PHOBIC' , 'CERBRO-VASCULAR DISEASE PRESENT' , 'DEPRESSIVE ILLNESS' ,
    'DIAGNOSTIC CODE' , 'EST OF SEVERITY OF DEPRESSION' , 'EST SEVERITY OF DEMENTIA' , 'PRIMARY PSYCHIATRIC DIAGNOSES' , 'AGE
    LEFT SCHOOL' , 'MOCA: YEARS OF EDUCATION' , 'YEARS IN FURTHER EDUCATION' , 'HISTORY OF STROKE' , 'MEDICAL ASSESSMENT V
    2010: STROKE' , 'HACHINSKI ISCHAEMIC: HISTORY OF STROKE' , 'GENERAL INFORMATION: DIABETES: DURATION' , 'GENERAL
    INFORMATION: DIABETES' , 'FEELING DEPRESSED' , 'DEPRESSED MOOD' , 'SEVERITY OF DEPRESSION' , 'MEDICAL ASSESSMENT V 2010:
    DEPRESSION' , 'MEDICAL ASSESSMENT V 2010: DEPRESSION TREATED BY DOCTOR' , 'HACHINSKI ISCHAEMIC: DEPRESSIVE
    SYMPTOMATOLOGY' , 'NPI: DEPRESSION/DYSPHORIA: FREQUENCY' , 'NPI: DEPRESSION/DYSPHORIA: SEVERITY' , 'NPI:
    DEPRESSION/DYSPHORIA: DISTRESS' , 'SPECT SCAN: DIAGNOSTIC ASSESSMENT' , 'BIOCHEMISTRY: CHOLESTEROL' , 'BIOCHEMISTRY: HDL
    CHOLESTEROL' , 'BIOCHEMISTRY: CHOLESTEROL/HDL RATIO' , 'HISTORY OF HEAD INJURY' , 'MEDICAL ASSESSMENT V 2010: HEAD
    INJURY' , 'SMOKING: SMOKING' , 'SMOKING: AVERAGE PER WEEK' , 'SMOKING: PIPES OR CIGARS' , 'SMOKING: CIGARETTES' , 'SMOKING:
    CIGARETTES: NO. PER YEAR' , 'SMOKING: TWO YEARS AGO' , 'SMOKING: TEN YEARS AGO' , 'SMOKING: TWENTY YEARS AGO' , 'SMOKING:
    THIRTY YEARS AGO' , 'SMOKING: AGE STARTED CIGARETTES' , 'SMOKING: TEA PER DAY' , 'SMOKING: COFFEE PER DAY' , 'SMOKING:
    ALCOHOL IN PAST 12 MONTHS' , 'SMOKING: COMPARED TO 5 YEARS AGO' , 'SMOKING: NON-DRINKER ALMOST ALWAYS' , 'MEDICAL
    ASSESSMENT V 2010: AGE STOPPED TOBACCO' , 'SHORTENED CAMBRIDGE ADL: FORGET TO PASS ON PHONE MESSAGES' , 'MEMORY PROBLEM' ,
    'CLINICAL DEMENTIA RATING: MEMORY' , 'DURATION OF MEMORY PROBLEMS' , 'ONSET OF MEMORY PROBLEMS' , 'CHANGE IN MEMORY
    PROBLEMS' , 'GDS: MEMORY PROBLEMS' , 'CAMCOG REMOTE MEMORY SCORE' , 'CAMCOG RECENT MEMORY SCORE' , 'CAMCOG LEARNING MEMORY
    SCORE' , 'MEDICAL ASSESSMENT V 2010: MEMORY' , 'SHORTENED CAMBRIDGE ADL: DIFFICULTY WITH MEMORY' , 'SHORTENED CAMBRIDGE
    ADL: POOR DAY-TO-DAY MEMORY' , 'MEDICAL ASSESSMENT V 2010: ALCOHOL CONSUPTION' , 'MEDICAL ASSESSMENT V 2010: AGE STOPPED
    ALCOHOL' , 'COGNITIVE EXAM 120-161: COGNITIVE EXAM 120-161' , 'IDENTIFIES DAY OF WEEK' , 'IDENTIFIES DATE' , 'IDENTIFIES
    MONTH' , 'IDENTIFIES YEAR' , 'IDENTIFIES SEASON' , 'IDENTIFIES COUNTY' , 'IDENTIFIES TOWN' , 'IDENTIFIES STREETS/COUNTRY' ,
    'IDENTIFIES FLOOR' , 'IDENTIFIES PRESENT PLACE' , 'COMPREHENDS NOD' , 'COMPREHENDS TOUCH' , 'COMPREHENDS LOOK' ,
    'COMPREHENDS TAP' , 'COMPREHENDS HOTEL' , 'COMPREHENDS VILLAGE' , 'COMPREHENDS RADIO' , 'IDENTIFIES OBJECTS' , 'IDENTIFIES
    OBJECTS: PENCIL' , 'IDENTIFIES OBJECTS: WATCH' , 'NAMES PICTURES' , 'NAMES PICTURES: SHOE' , 'NAMES PICTURES: TYPEWRITER' ,
'NAMES PICTURES: SCALES' , 'NAMES PICTURES: SUITCASE' , 'NAMES PICTURES: BAROMETER' , 'NAMES PICTURES: LAMP' , 'NUMBER OF
    ANIMALS LISTED' , 'NUMBER OF ANIMALS LISTED: SCORE' , 'DEFINES HAMMER' , 'REPETITION' , 'RECALLS OBJECTS' , 'RECALLS
    OBJECTS: SHOE' , 'RECALLS OBJECTS: TYPEWRITER' , 'RECALLS OBJECTS: SCALES' , 'RECALLS OBJECTS: SUITCASE' , 'RECALLS
```

OBJECTS: BAROMETER', 'RECALLS OBJECTS: LAMP', 'RECOGNISES PICTURES: SHOE', 'RECOGNISES PICTURES: SCALES', 'RECOGNISES PICTURES: BAROMETER', 'REMEMBERS WW1 DATE', 'REMEMBERS WW2 DATE', 'REMEMBERS HITLER', 'REMEMBERS STALIN', 'REMEMBERS MAE WEST', 'REMEMBERS LINDBERGH', 'KNOWS MONARCH', 'KNOWS HEIR TO THRONE', 'KNOWS PRIME MINISTER', 'KNOWS RECENT NEWS ITEM', 'REGISTERS OBJECTS', 'REGISTERS OBJECTS 1: APPLE', 'REGISTERS OBJECTS 3: PENNY', 'REGISTERS OBJECTS: REPEATS', 'COUNTING BACKWARDS', 'SPELL BACKWARD', 'RECALLS OBJECTS', 'RECALLS OBJECTS 1: APPLE', 'RECALLS OBJECTS 2: TABLE', 'RECALLS OBJECTS 3: PENNY', 'COGNITIVE EXAM 162–187: COGNITIVE EXAM 162–187', 'READING COMPREHENSION 1', 'READING COMPREHENSION 2', 'DRAWS PENTAGON', 'DRAWS SPIRAL', 'DRAWS HOUSE', 'CLOCK DRAWING', 'CLOCK DRAWING: CIRCLE', 'CLOCK DRAWING: NUMBERS', 'CLOCK DRAWING: TIME', 'WRITES A SENTENCE', 'PRAXIS –PAPER', 'PRAXIS – PAPER: RIGHT HAND', 'PRAXIS – PAPER: FOLDS', 'PRAXIS – PAPER: ON LAP', 'PRAXIS – ENVELOPE', 'DICTATION', 'MIME – WAVE', 'MIME – SCISSORS', 'MIME – BRUSHING TEETH', 'IDENTIFIES COIN', 'ADDS UP MONEY', 'SUBTRACTS MONEY', 'RECALLS ADDRESS', 'RECALLS ADDRESS: JOHN', 'RECALLS ADDRESS: BROWN', 'RECALLS ADDRESS: D42', 'RECALLS ADDRESS: WEST', 'RECALLS ADDRESS: BEDFORD', 'SIMILARITIES – FRUIT', 'SIMILARITIES – CLOTHING', 'SIMILARITIES – FURNITURE', 'SIMILARITIES – LIFE', 'RECOGNISES FAMOUS PEOPLE', 'RECOGNISES OBJECTS', 'RECOGNISES OBJECTS: SPECTACLES', 'RECOGNISES OBJECTS: SHOE', 'RECOGNISES OBJECTS: PURSE', 'RECOGNISES OBJECTS: CUP', 'RECOGNISES OBJECTS: TELEPHONE', 'RECOGNISES OBJECTS: PIPE', 'RECOGNISE PERSON', 'PATIENT', 'COGNITIVE EXAM 162–187: HANDED', 'COGNITIVE IMPAIRMENT', 'PHYSICAL SYMPTOMS', 'PHYSICAL PROBLEMS', 'PHYSICAL EXAM 213–234: PHYSICAL EXAM 213–234', 'BLOOD PRESSURE', 'BLOOD PRESSURE: SYSTOLIC', 'BLOOD PRESSURE: DIASTOLIC', 'TENDON REFLEXES', 'PLANTAR REFLEXES', 'HEMIPARESIS', 'GAIT', 'MOBILITY', 'DEAFNESS', 'VISUAL DEFECT', 'TREMOR', 'MANUAL DIFFICULTY', 'ABNORMAL EYE MOVEMENTS', 'SHORTNESS OF BREATH', 'FULL BLOOD COUNT', 'B12 OR FOLATE', 'THYROID FUNCTION TESTS', 'UREA AND ELECTROLYTES', 'SKULL XRAY OR SPECT SCAN', 'LIVER FUNCTION TESTS', 'CT OR MRI SCAN', 'VDRL', 'CAUSES OF DEMENTIA EXCLUDED', 'PHYSICAL EXAM 213–234: SUBJECT ON MEDICATION', 'GDS: AVOID SOCIAL GATHERINGS', 'APOE', 'APOE: RESULT', 'ANXIOUS', 'ANXIOUS SITUATIONS', 'ANXIOUS OR FEARFUL', 'MEDICAL ASSESSMENT V 2010: ANXIETY', 'NPI: ANXIETY: FREQUENCY', 'NPI: ANXIETY: SEVERITY', 'NPI: ANXIETY: F X S', 'NPI: ANXIETY: DISTRESS', 'GENERAL INFORMATION: ASPIRIN: DURATION', 'GENERAL INFORMATION: ASPIRIN', 'SMOKER', 'EDUCATION', 'AGE', 'GENDER', 'APOE', DURATION(years)']

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
