# Peer review of "Calibrating Mini-Mental State Examination Scores to Predict Misdiagnosed Dementia Patients"

_applsci, doi:10.3390/app11178055_

Round 1

Reviewer 1 Report

This study is well organized overall and seems to be an interesting topic with significant results.
I have listed some developable points as follows:

 1. The abstract is well written and express about the purpose of this research. The authors can add the random forest prediction model as a keyword.

2. There should be reference for 1st part of the introduction.

3. The literature review was written about MMSE only. You need to add more information about "Machine learning", and "Classification" to support your research to be significant and accurate for the conclusion.

4. Overall, there is too little reference in the introduction and theoretical background. Therefore, it is questionable whether the research methods and topics brought by the studies need to be studied, eliminating such doubts and citing more significant references for strong introduction and theoretical background.

5. There is not enough information to understand the whole research method cause there is no literature review for that or not enough explanation in the methodology part.

6. What is OPTIMA dataset? Is that reliable data source? Please prove the is reliable and good enough use in the paper.

7. What is the random forests algorithm and why you have adopted that algorithm?

8. How this data can be the qualitative and quantitative analysis?

9. Not enough information regarding to the 4.2. dataset cleaning and preprocessing.

Thank you.

Author Response

Response to Reviewer 1 Comments

Point 1: The abstract is well written and express about the purpose of this research. The authors can add the random forest prediction model as a keyword.

Response 1: We've included "Random forest" and "Predictive models" to our keyword list as a result of your recommendation.

Point 2: There should be reference for 1st part of the introduction.

Response 2: There is now a reference.

Point 3: The literature review was written about MMSE only. You need to add more information about "Machine learning", and "Classification" to support your research to be significant and accurate for the conclusion.

Response 3:  In the last three paragraphs, we have broadened our literature review (2) and included more facts.

Point 4: Overall, there is too little reference in the introduction and theoretical background. Therefore, it is questionable whether the research methods and topics brought by the studies need to be studied, eliminating such doubts and citing more significant references for strong introduction and theoretical background.

Response 4:  More references could be found in the introduction (1), literature review (2), problem statement (3.1), predictive models (3.2.2), and dataset sections (4.1). Furthermore, we have attempted to make these portions more understandable.

Point 5: There is not enough information to understand the whole research method cause there is no literature review for that or not enough explanation in the methodology part.

Response 5:  In the last three paragraphs, we have broadened our literature review (2) and included more facts.

Point 6: What is OPTIMA dataset? Is that reliable data source? Please prove the is reliable and good enough use in the paper.

Response 6: In section dataset (4.1), we've added more references and details about the OPTIMA dataset.

Point 7: What is the random forests algorithm and why you have adopted that algorithm?

Response 7:  In the last paragraph of the literature review (2) and predictive models (3.2.2), we attempted to respond to this comment.

Point 8: How this data can be the qualitative and quantitative analysis?

Response 8: In section dataset (4.1), we've added more references and details about the OPTIMA dataset.

Point 9: Not enough information regarding to the 4.2. dataset cleaning and preprocessing.

Response 9: To make data cleaning & preprocessing (4.2) better, more information has been provided.

Reviewer 2 Report

1.

This reference is wrong [8]. The paper that is referred is not the good one (not about OPTIMA): Clarke, R.; Smith, A.D.; Jobst, K.A.; Refsum, H.; Sutton, L.; Ueland, P.M. Folate, vitamin B12, and serum total homocysteine levels

in confirmed Alzheimer disease. Archives of neurology 1998, 55, 1449–1455.

All the other references have to be checked.

3.1 problem statement

Many references are missing. References should be stated to each statement.

3.2.2

Why Random Forests algorithm is used instead of using other classification algorithm such as C4.5 for example?

Why only two predictive models are used  ?

3.3

Justify the use of the chosen performance measures.

How will you approach deal with the new cases that do not exist in the used dataset for training?  It is not mentioned.

4.2

How these ordinal encoding (e.g. size: Small, Medium, Large) were applied?

5.

Discussion section must be before the conclusions one. The way the section is presented is like conclusions are being discussed while it is the results that are discussed.

How this approach will be used in a real clinical decision support system?

How about the limits of this work?

Author Response

Response to Reviewer 2 Comments

1.

Point 1: This reference is wrong [8]. The paper that is referred is not the good one (not about OPTIMA): Clarke, R.; Smith, A.D.; Jobst, K.A.; Refsum, H.; Sutton, L.; Ueland, P.M. Folate, vitamin B12, and serum total homocysteine levels in confirmed Alzheimer disease. Archives of neurology 1998, 55, 1449–1455.

Response 1: This has been revised.

Point 2: All the other references have to be checked.

Response 2: This has been revised.

3.1 problem statement

Point 3: Many references are missing. References should be stated to each statement.

Response 3: We've included more references and attempted to make it better (problem statement (3.1)).

3.2.2

Point 4: Why Random Forests algorithm is used instead of using other classification algorithm such as C4.5 for example?

Response 4: In the last paragraph of the literature review (2) and predictive models (3.2.2), we attempted to respond to this comment.

Point 5: Why only two predictive models are used ?

Response 5: In the last paragraph of the literature review (2) and predictive models (3.2.2), we attempted to respond to this comment.

3.3

Point 6: Justify the use of the chosen performance measures.

Response 6: More information was added to clarify the performance measures (Evaluation metrics(3.3)).

Point 7: How will you approach deal with the new cases that do not exist in the used dataset for training?  It is not mentioned.

Response 7: In the discussion section (5), we attempted to respond.

4.2

Point 8: How these ordinal encoding (e.g. size: Small, Medium, Large) were applied?

Response 8: More information was added to clarify it (Data cleaning & Preprocessing (4.2)).

5.

Point 9: Discussion section must be before the conclusions one. The way the section is presented is like conclusions are being discussed while it is the results that are discussed.

Response 9: For discussion and conclusions, we've created a separate section.

Point 10: How this approach will be used in a real clinical decision support system?

Response 10: In the discussion section (5) - paragraph -4, we attempted to respond.

Point 11: How about the limits of this work?

Response 11: In the discussion (5) and conclusions (6) sections, we attempted to respond.